# Construction of Carbon Microspheres-Based Silane Melamine Phosphate Hybrids for Flame Retardant Poly(ethylene Terephthalate)

**DOI:** 10.3390/polym11030545

**Published:** 2019-03-22

**Authors:** Baoxia Xue, Ruihong Qin, Jie Wang, Mei Niu, Yongzhen Yang, Xuguang Liu

**Affiliations:** 1College of Textile Engineering, Taiyuan University of Technology, Yuci 030600, China; 18834075502@163.com (R.Q.); 18835170553@163.com (J.W.); 2Key Laboratory of Interface Science and Engineering in Advanced Materials, Taiyuan University of Technology, Ministry of Education, Taiyuan 030024, China; yyztyut@126.com (Y.Y.); liuxuguang@tyut.edu.cn (X.L.); 3Research Center on Advanced Materials Science and Technology, Taiyuan University of Technology, Taiyuan 030024, China; 4College of Material Science and Engineering, Taiyuan University of Technology, Taiyuan 030024, China

**Keywords:** carbon microspheres, melamine polyphosphate, smoke suppression, flame retardant, poly(ethylene terephthalate)

## Abstract

To improve the flame retardancy and inhibit the smoke of poly(ethylene terephthalate) (PET), carbon microspheres (CMSs)-based melamine phosphate (MP) hybrids (MP-CMSs) were constructed in situ with the introduction of CMSs into the hydrothermal reaction system of MP. The integrated MP-CMSs were modified by 3-Aminopropyltriethoxysilane (APTS) to obtain the silane MP-CMSs (SiMP-CMSs) to strengthen the interface binding between the MP-CMSs and PET matrix. The results showed that the SiMP layer was loaded on the CMSs surface. The addition of only 3% SiMP-CMSs increased the limiting oxygen index (LOI) value of the PET from 21% ± 0.1% to 27.7% ± 0.3%, reaching a V-0 burning rate. The SiMP-CMSs not only reduced heat damage, but also inhibited the smoke release during PET combustion, whereupon the peak heat release rate (pk-HRR) reduced from 513.2 to 221.7 kW/m^2^, and the smoke parameters (SP) decreased from 229830.2 to 81892.3 kW/kg. The fire performance index (FPI) rose from 0.07 m^2^s/kW to 0.17 m^2^s/kW, demonstrating the lower fire risk. The proportion of the flame-retardant mode in the physical barrier, flame inhibition, and char effects were recorded as 44.53%, 19.04%, and 9.04%, respectively.

## 1. Introduction

Poly(ethylene terephthalate) (PET) is one of five major general-purpose plastics with excellent mechanical properties, fiber formation characteristics, and wear resistance [1]. PET has widely been used in engineering plastics, fiber, film, packaging, and other fields [2]. However, PET is combustible, producing significant black smoke during combustion. On the other hand, smoke, which can cause suffocation, is considered a fatal factor in fire. Smoke suppression and flame retardancy are equally important, but the smoke harm is often ignored. Therefore, achieving flame retardancy and smoke suppression for PET at the same time is highly desirable.

The intumescent flame-retardant system (IFR) has low smoke and low toxicity, and is, therefore, considered to be an environmentally friendly flame-retardant system. However, high amounts of IFR are added to meet flame retardant requirements. IFR is mainly composed of a carbon source, acid, and gas. The carbon source is rich in hydroxyl groups, such as pentaerythritol (PER) [3,4]. However, the traditional carbon source exists in defects of low molecular weight, which could migrate easily to deteriorate the mechanical properties of a polymer [3]. Substantial research has recently been devoted to the development of new charring agents, which are mainly based on natural polymers rich in hydroxyl groups, such as cyclodextrin [5], chitosan [6], and lignin [7]. However, due to the low decomposition temperature, the charring agent could not satisfy the process temperature for PET (300 °C). Carbon material flame retardant systems are new and promising systems for polymers, due to their efficient thermal stabilities. These systems include carbon nanotubes, carbon black, and graphene, among others. The low content (0.5 wt %–5 wt %) of carbon materials can obviously reduce the heat release rate of polymers during combustion. Thus, the addition of carbon materials in fire-retardant systems can reduce the amount of flame retardant for PET. As synergists in IFR systems, Carbon materials are often used for a variety of purposes, such as the synergistic effect of expandable graphite on the melamine polyphosphate/dipentaerythritol/polypropylene (PP) [8], multiwalled carbon nanotubes functionalized with IFR on improving the flame retardant of the ABS matrix [9], thermally reduced graphite oxide strengthening the residue structure of ammonium polyphosphate /magnesium hydroxide/PP [10], et al. Therefore, carbon materials could replace traditional carbon sources to avoid their defects and provide carbon materials as flame retardant polymers.

Compared to other carbon materials, carbon material-carbon microspheres (CMSs) represent a low degree graphitization of fullerene with abundant hydroxyl groups on their surfaces. These features provide CMSs with traditional carbon sources of IFR systems and the advantages of carbon materials as flame retardant. According to our previous study [11], CMSs begin to decompose at 376 °C, satisfying the process requirement for PET and able to catalyze the PET to char. Hence, as a new carbon source, CMSs are expected to play the role of char agent.

To achieve low smoke generation and high flame retardancy, instead of a smoke suppressant, the acid source and gas sources were introduced onto the CMSs surface to form integrative CMSs-based composite flame retardants with “three sources in one” by in situ modification. Compared to conventional compounding—the direct melt blending method—integrative composite flame retardants could not only prevent interface problems between every filler and polymer matrix but also play a comprehensive role in achieving highly efficient flame retardancy. As the phosphorus element is the most effective flame-retardant element for PET, phosphoric acid was used as an acid source. Melamine (MEL) is not only rich in amino groups, but also has a triazine structure. As a gas source, it has widely been used in the IFR system [12,13,14,15]. At present, a series of derivatives based on MEL with a triazine structure have been prepared, and the synergistic flame retardancy between MEL and some carbon materials has been demonstrated. These include expandable graphite [8], graphene [13], and carbon nanotubes [14]. Therefore, the MEL was selected as the gas source.

Compared with our previous work [16,17], another way to solve flame retardant and smoke suppression simultaneously was stated in this study. Integrated CMSs-based melamine phosphate (MP) composite flame retardant (MP-CMSs) was in-situ prepared by the solvent hydrothermal method. To prevent interface problems between the MP-CMSs and the PET matrix, the MP-CMSs composites were further modified with 3-Aminopropyltriethoxysilane (APTS) to obtain silane MP-CMSs (SiMP-CMSs). The SiMP-CMSs/PET samples were prepared by a melt blending method. The effects of the SiMP-CMSs on flame retardant properties and the smoke suppression of PET composites were systematically investigated. The flame-retardant mode was quantitatively calculated and the mechanism was examined.

## 2. Materials and Methods

### 2.1. Materials

CMSs (sized ~600 nm in diameter) were synthesized by the hydrothermal method. Glucose (C_6_H_12_O_6_), melamine (MEL), sodium dodecyl benzene sulfonate (SDBS), *N*,*N*-two methyl formamide (DMF), absolute ethanol, phosphoric acid (H_3_PO_4_), *N*,*N*’-Dicyclohexylcarbodiimide (DCC), and benzene were all analytical reagent (AR) grade and obtained from Beijing Chemical works, Beijing, China. 3-Aminopropyltriethoxysilane (APTS) was purchased from Sigma-Aldrich (Shanghai, China) and PET pellets (SD500) with a viscosity index of 0.68 dL/g from Sinopec Yizheng, Jiangsu, China.

### 2.2. In Situ Construction of SiMP-CMSs Composites

**MEL modified CMSs:** A mixture of 100 mL ethanol/distilled water (1:1) was added to a three flasks container. Next, 0.5 g CMSs, 0.6 g MEL, and 0.09 g DCC were added to the solvent mixture. After ultrasonic agitation for 30 min, the mixture was heated to 80 °C and maintained for 18 h, then cooled to room temperature. The reaction product was filtered off with absolute ethanol and hot water to remove unreacted MEL. The final product was then dried in an oven at 80 °C for 5 h to obtain MEL-CMSs powder.

**Construction of SiMP-CMSs composites**: Using 70 mL benzene as solvent, 0.5 g MEL-CMSs, 0.1 g MEL and 0.015 g SEBS were added as additives and the mixture was compounded as homogeneous solution. Under stirring, 3 mL H_3_PO_4_ was added to the solution at constant speed using a separating funnel. The mixture was then sealed in a Teflon reactor of 100 mL. At 150 °C, the Teflon reactor was heated for 3 h, and then stirred and cooled down to room temperature. The obtained reaction product was filtered with DMF, absolute ethanol and hot water to remove unreacted reagents. The final product was dried in an oven at 110 °C for 5 h to obtain the MP-CMSs powder.

A three-necked flask was filled with 1 g MP-CMSs and 100 mL absolute ethanol. Afterward, 0.1 mL APTS was poured into 10 mL distilled water and sonicated for 30 min. The APTS solution was then added to the three-necked flask, and the mixture was left to react at 80 °C for 7 h under condensation conditions. The resulting solids were washed, filtrated off, and dried under a vacuum to yield the products defined by SiMP-CMSs.

### 2.3. Preparation of SiMP-CMSs/PET Composites

To remove moisture from the material, the SiMP-CMSs powder and PET pellets were firstly dried in a vacuum oven at 120 °C for 8 h before use, and then added into a two-screw extruder (CET35–40D, Kobelong Machinery Co., Ltd., Nanjing, China). The temperature of the screw section varied from 262–270 °C at a screw rotation speed of 180 rpm. Next, the extrudates were cooled with cold water, cut into pellets, and the resulting compounds were named SiMP-CMSs/PET. Finally, the pellets were processed by the injection-molding (52–80ONB-Aingbo Plastics Machinery, China) process at temperatures varying from 265–275 °C for testing flame retardancy and mechanical properties.

### 2.4. Materials Characterization

Field-emission scanning electron microscopy (FESEM; JSM-6700F, JEOL, Tokyo, Japan) with X-ray energy dispersive spectroscopy (EDS) were used to examine the morphologies of the different samples. Fourier transformation infrared (FTIR) spectra were recorded using a Tensor 27 spectrometer (Bruker) in the infrared domain between 4000 and 500 cm^−1^ at a resolution of 4 cm^−1^. The chemical structures of the samples were examined by X-ray diffraction (XRD) with a RigakuD/MAX-2500 V/PV using Cu Kα radiation (40 kV and 200 mA) at a scanning speed of 0.5°/min over the 2θ range of 10°–80°. The thermal stability and mass percentage of both the fillers and PET composites were investigated using a Netzsch TG209F3 instrument at a heating rate of 10℃/min under a nitrogen atmosphere. Cone calorimeter measurements were performed on an FTT Cone calorimeter according to ISO5660 under an external heat flux of 50 kW/m^2^. The time to ignition (TTI), total heat release (THR), mean effective heat of combustion (MEHC), smoke production rate (SPR), total smoke production (TSP), and mean specific extinction area (av-SEA) parameters were directly obtained from the cone results. Fire performance index (FPI) (FPI = TTI/pk-heat release rate (HRR)) and the smoke parameter (SP) (SP = av-SEA*pk-HRR) were calculated from the cone results. The limiting oxygen index (LOI) values were measured by means of a TM606 oxygen index meter obtained from Qing-dao Analyzing Instrument Company (Qingdao, China). This measurement was performed on specimens of 130 mm × 6.5 mm × 3 mm according to the standard oxygen index test ASTM D2863-97. The UL-94 vertical burning tests were executed using a CZF-5 type instrument (Shanfang Instrument CO., LTD, Qingdao, China) on sheets of 125 mm × 13 mm × 13 mm following the standard UL-94 test ASTM D 3801-2010. Pyrolysis-gas chromatography-mass spectrometry (Py-GC-MS) experiments were carried out using a system consisting of a FRONTIER PY-2020iD pyrolyser and a GCMS QP2010 (Shinadzu CO., Kyoto, Japan). Testing conditions were as follows: carrier gas speed: 0.8 mL (He)/min; injector temperature: 250 °C; pyrolysis temperature: 600 °C; the column was kept at 60 °C for 5 min and heated up to 250 °C at a rate of 8 °C/min, and then to 280 °C at a rate of 6 °C /min and maintained at 250 °C for 5 min. The tensile test was carried out by a universal material testing system (model CMT6104, Shenzhen, China) at room temperature with a gauge length of 30 mm and a crosshead speed of 50 mm/min.

## 3. Results and Discussion

### 3.1. Morphology and Chemical Structure of SiMP-CMSs

The microstructure of CMSs and SiMP-CMSs hybrids were characterized by SEM, and the results are gathered in Figure 1. CMSs showed a spherical shape with smooth morphology of the outer surface (Figure 1a). By comparison, Figure 1b revealed that the SiMP-CMSs was rougher, indicating that new material formed on the outer surface of the CMSs. The EDS spectra of SiMP-CMSs displayed the presence of C, O, N, P, and Si elements on the surface (Figure 1b), demonstrating that SiMP material may exist on the CMSs’ surface.

The chemical structure of CMSs and SiMP-CMSs were investigated by FTIR and XRD. The FTIR profiles depicted bands at 3315 cm^−1^ and 1226 cm^−1^ for the CMSs (Figure 2a), assigned respectively to O–H stretching vibration and C–OH bending vibration [18]. The bands at 1689 cm^−1^ and 1593 cm^−1^ were attributed to the C=O stretching vibration and bending vibration. These features indicated that O-H and C=O groups were the major functional groups for CMSs. For MP-CMSs, few new characteristic peaks appeared between 3400–2900 cm^−1^ and 1600–700 cm^-1^ (Figure 2b). The different vibration form of the triazine ring skeleton (1500, 1454 cm^−1^), amino (3211, 3094 cm^−1^), P=O (1271 cm^−1^), P–O (1086, 1047 cm^−1^), and P–OH (970 cm^−1^) were all characteristic absorption peaks of MP [6]. In addition, the C=O of CMSs shifted to low frequencies (1674, 1544 cm^−1^), mainly due to the direct connection between C=O and N atoms. This process yielded a conjugation effect, which was stronger than the induction effect with N atoms that weakened the properties of C=O [15]. This result demonstrated that the C=O of CMSs was combined with the NH_2_ of MP through condensation forces. The bending vibration peak of C=N was superimposed with that of CO–NH at 1674 cm^−1^, enhancing the vibration strength.

After the modification of the MP-CMSs by APTS (Figure 2c), the N–H strength vibration peak of APTS was visible at 2974 cm^−1^. The two peaks at 1086 cm^−1^ and 1047 cm^−1^ moved to a high frequency (1097, 1055 cm^−1^). These features were mainly due to the superposition of the Si–O–C and Si–O–Si bond vibration absorption peak of APTS [19]. These features implied that MP-CMSs and APTS were mainly connected by dehydration condensation between Si–OH and P–OH. From the above analyses, the formation process of SiMP-CMSs can be summarized, as in Figure 3.

To further analyze the structural changes of the CMSs, both the CMSs and the SiMP-CMSs were tested by XRD, and the results are shown in Figure 4. The diffraction peak in the XRD pattern of the CMSs appeared at 24.5°, corresponding to the (002) reflection of the graphite. The diffraction curve of the MP in Figure 4b revealed five main diffraction peaks for neat MP. The peaks were observed at 2θ of 13.5°, 20.2°, 27.9°, 30.3°, and 36°, corresponding respectively to the (110), (200), (101), (220), and (030) crystal face diffraction peaks of MP [15]. Compared to the neat CMSs and MP, the characteristic diffraction peaks of the CMSs and MP appeared at the same position in the XRD profiles of the SiMP-CMSs (Figure 4c). These data further demonstrated that SiMP can successfully be formed on CMSs.

The thermal stability and weight ratios were investigated. The TG and DTG curves of neat CMSs, neat SiMP, and SiMP-CMSs are gathered in Figure 5. Neat CMSs began to decompose at 376 °C, showing about a 5% weight loss. From the DTG curve, there was only one maximum weightlessness peak at about 570 °C for the CMSs. The final char residue at 700 °C was about 64.38%. The initial decomposition temperature of the SiMP was recorded at 355 °C, lower than that of the CMSs by nearly 20 °C. Combined to the DTG curves, the SiMP turned out to be gradient decomposition. The first weight loss of about 40% occurred at 355–450 °C. The main products at this stage were identified as MEL and phosphoric acid. MEL further vaporized and decomposed to produce ammonia gas [20]. Moreover, melamine reacted with phosphoric acid by self-polymerization to form a thermal polymer, which would further decompose at a high temperature. Therefore, the first and second maximum weightlessness peak occurred at about 450 °C and 720 °C. The final residue of the SiMP layer at 700 °C was around 17.08%.

The SiMP-CMSs in Figure 5b decomposed in advance (at about 300 °C), mainly due to the decomposition of unstable reaction products on the surface of the SiMP-CMSs. Two turning points emerged at 350 and 430 °C on the TG curve, and two maximum weightlessness peaks occurred at 441 and 515 °C on the DTG curves of the SiMP-CMSs, indicating that the SiMP layer occurred on the surface of the CMSs. The char residue of the SiMP-CMSs at 700 °C was determined to about 34.04%, which was nearly two times higher than that of the SiMP. According to the final residue of the CMSs, SiMP and SiMP-CMSs, the mass ratio between SiMP and CMSs in the SiMP-CMSs was about 16 to 9. These results proved that the SiMP layer can load on the CMSs surface.

### 3.2. Flammability, Smoke, and Gas Production

The flammability properties of the PET composites were assessed by UL-94 and LOI tests, and the results are presented in Table 1. Pure PET was found combustible, with an LOI value of 21% and no classification in the UL-94 rating. The addition of 0.5%SiMP-CMSs allowed the SiMP-CMSs/PET to acquire a UL-94 V-2 grade with a LOI value of 25.8%. Moreover, the total burning time of SiMP-CMSs/PET decreased from 85.5 s to 56.5 s. As the SiMP-CMSs increased, the LOI value of the SiMP-CMSs/PET composites showed reversed trends. At a SiMP-CMSs addition of 3%, the SiMP-CMSs/PET composites passed the V-0 grade and reached an LOI value of 27.7% ± 0.3%. Moreover, the total burning time shortened to 23.5 s because of the heat barrier of SiMP-CMSs in this range and the vaporization of MELon the sample surface. This decomposition stage can produce NH_3_ and H_2_O vapor, rapidly diluting the O_2_ concentration and the heat quantity of the surrounding flame, as well as the phosphoric acid catalysis of PET [21]. These two aspects rose the LOI value, suppressed combustion, and shortened the total burning time.

The UL-94 rating of the PET composites changed as the SiMP-CMSs amount rose. The accumulation of SiMP-CMSs to certain amounts induced rapid dehydration by the decomposed products of MP and H_3_PO_4_ combined with carbonization and rapid cooling. This, in turn, extinguished the flames of the droplets without causing the combustion of the skimmed cotton.

To further assess the flame-retardant properties of the PET composites, the combustion heat was quantitatively analyzed by the cone method. The cone method can accurately measure all kinds of combustion parameters [22]. Figure 6 shows the HRR and THR curves of the SiMP-CMSs/PET composites. The corresponding combustion parameters are listed in Table 2. The addition of the SiMP-CMSs reduced the HRR and THR values during combustion. This reduction was mainly attributed to the phosphoric acid’s ability to catalyze the char formation of the PET in the condensed phase, forming an effective heat insulation carbon layer. The decrease in the MEHC value also indicated a reduction in either heat released flammable volatile components or the degree of flame combustion in the gas phase. The lowest MEHC value was estimated to be 18.6 MJ/kg. The reason for this measurement concerns two factors: (1) the addition of SiMP-CMSs reduced combustible volatile content by catalyzing the PET into char, and (2) the barrier effect of the carbon layer prevented the flammable volatiles from escaping. The reduction in heat-related parameters proved that the SiMP-CMSs could reduce the heat damage of PET combustion.

Furthermore, by comparing the residue data shown in Table 2, the residue of the SiMP-CMSs/PET composites appeared all higher than that of the PET, demonstrating that the SiMP-CMSs played a positive role in catalyzing the char of the PET. By adding 3% SiMP-CMSs, the char amount of the SiMP-CMSs/PET reached 19.8%, and the FPI value of the SiMP-CMSs/PET composites significantly improved. This indicated a decrease in fire risk. At SiMP-CMSs additions of 0.5%, 1%, 2% and 3% increased the FPI value of the SiMP-CMSs/PET composites from 0.07 m^2^s/kW to 0.12, 0.13, 0.15, and 0.17 m^2^s/kW.

Compared to traditional testing technologies, the cone method can also measure the dynamic generation of smoke during combustion. The SPR and TSP curves are shown in Figure 7 and the corresponding specific parameters are listed in Table 3.

The SPR and TSP curves in Figure 7 showed that the addition of SiMP-CMSs reduced the smoke production rate and total smoke production during the combustion process of PET. As SiMP-CMSs rose, the TSP, av-SEA value, and CO and CO_2_ yield of the SiMP-CMSs/PET composites all declined. These results suggest that the smoke and gas release amounts decreased. The corresponding lowest values were recorded as 12.1 m^2^, 369.5 m^2^/kg, 0.04 kg/kg, and 1.03 kg/kg. However, the av-SEA parameter did not show the influence of pk-HRR. In real fire situations, the material with a high SEA and low HRR value may not burn or extinguish quickly enough to stop producing smoke. However, materials with low SEA and high HRR may cause fire and produce much smoke. To simultaneously measure SEA and HRR, the SP was introduced to assess the smoke generating capacity of SiMP-CMSs/PET composites during the combustion process. The smaller value would induce a lower ability to produce smoke [22]. The combination as a function of SP in Table 3 indicates that the addition of the SiMP-CMSs obviously reduced the SP value of the PET. At SiMP-CMSs/PET amounts of 0.5%, 1%, 2%, and 3%, the SP values of the SiMP-CMSs/PET composites decreased by 40.71%, 47.65%, 57.24%, and 64.37%, when compared to pure PET (229830.2 kW/kg). The reasons for the decline in the smoke capacity of the SiMP-CMSs/PET composites had mainly to do with two factors. First, the MP loaded on the CMS surface decomposed the MEL and H_3_PO_4_, and then the MEL vaporized to escape the material surface. The decomposition releasing NH_3_ induced dilution of the O_2_ concentration and prevented the thermal decomposition of the PET. Second, the H_3_PO_4_ can effectively catalyze the PET matrix carbonization, where the triazine ring structure of the MEL decomposition promoted crosslinking of the PET. This increased the char effect of the PET in the solid phase and rose the char amount.

### 3.3. Quantitative Analyses of Flame-Retardant Mode

Section 3.2 indicates that the addition of 3% SiMP-CMSs in the PET matrix increased the LOI value to 27.7% ± 0.3% and improved the UL-94 rating to the V-0 level. The Pk-HRR and THR values were respectively recorded as 221.7 kW/m^2^ and 56.0 MJ/m^2^, corresponding to a decrease of 56.8% and 22.1%, respectively. The TSP and SP were 12.1 m^2^ and 81892.3kW/kg, corresponding to reductions of 2.59 m^2^ and 147937.9 kW/kg, respectively. These data demonstrate the achievement of the double effect of flame retardant and smoke suppression.

To further investigate the specific flame-retardant mode, the results were quantified according to Equations (1)–(3) [23], and the results are gathered in Table 4. The barrier effect was mainly induced by the addition of the CMSs together.
E_Barrier_ = 1 − (pk-HRR_FRPET_/pk-HRR_PET_)/(THR_FRPET_/THR_PET_)(1)
E_Flame inhibition_ = 1 − MEHC_FRPET_/MEHC_PET_(2)
E_charring effect_ = 1 − TML_FRPET_/TML_PET_(3)

The barrier, flame inhibition and char effects were three flame-retardant modes affecting the SiMP-CMSs/PET composites. The proportion of barrier, flame inhibition, and char effects of the SiMP-CMSs/PET composites at 3% were estimated to be 44.53%, 19.04%, and 9.04%, respectively. In our previous work [11], the barrier effect of the CMSs for the PET was demonstrated, where the barrier effect of SiMP-CMSs was further strengthened due to the presence of a P=O bond. The P=O bond could reduce the permeability of the carbon layer to improve the barrier’s properties. The flame inhibition effect was mainly induced during the gasification and decomposition of MEL, where large amounts of NH_3_ diluted the O_2_ concentration. The charring effect was mainly seen on the catalytic charring of H_3_PO_4_ and promoting the crosslinking of a triazine ring. The three aspects of these comprehensive effects prevented the thermal stability of the PET and extinguished the flame in advance, achieving a flame retardant and smoke suppression.

### 3.4. Thermal Degradation Behavior, Volatilized, and Solid Phase Product

The thermal degradation of the SiMP-CMSs/PET composites (The mass fraction of the SiMP-CMSs was 1%) in air atmospheres was analyzed by TGA and DTG, and the results are shown in Figure 8 and Table 5. The thermal oxygen degradation behavior of the SiMP-CMSs/PET composites was divided into two stages. As shown in Table 5, the *T*_onest_ of the SiMP-CMSs/PET composites was delayed by nearly 10 °C. The *T*_max1_ and *T*_max2_ during the two stages were, respectively, 433.5 °C and 572.3 °C, all higher than those of the PET. Especially during the second stage, *T*_max2_ was delayed nearly by 37 °C. This result demonstrated that the thermal oxygen resistance of the carbon layer formed during later periods was obviously improved, mainly due to the covering of the triazine ring structure on the carbon layer and the bonding of the P=O bond to the carbon layer to induce passivation and ultimately improve thermal oxygen stability.

To investigate the thermal degradation mechanism, Py-GC-MS were selected to monitor the gaseous phase of the PET composites. The thermal decomposition of the PET and SiMP-CMSs/PET were qualitatively analyzed at 600 °C by Py-GC-MS, and the data are listed in Table 6. The mass fraction of the SiMP-CMSs in the PET matrix was 1%. The main pyrolysis products of pure PET were determined to be 34.01% benzene acid, 10.92% diethylene glycol dibenzoate, 8.05% alkyl benzoic acid, 7.25% phenylglyoxal, 7.23% benzene, 5.52% 4-biphenyl formalin, 4.15% 4-acetylbenzoic acid, 4.10% acetic acid, 3.03% formic acid, 2.64% 4-biphenyl formic acid, and 2.14% propanediol. The valence bond in the PET molecular chain mainly contained C–C, C–H, C–O, and C=O bonds, and the corresponding bonds’ energies were 80.65, 98.9, 78.0, and 163.2 KJ/mol [24], respectively. Therefore, the weakest bond was the C–O bond, namely β-methylene. The PET was first decomposed by a classical ester scission reaction to generate carboxylic acid and olefinic end groups, which were further converted via ester scission, rearrangement, and radical reactions.

Table 6 revealed some changes in the relative concentrations and compositions of pyrolysis products after the addition of the SiMP-CMSs. The main pyrolysis products of the SiMP-CMSs/PET composites were 38.57% benzene acid, 9.89 % diethylene glycol dibenzoate, 9.02% formic acid, 7.11% benzoyl methyl ketone, 5.84% 4-acetylbenzoic acid, 5.17% benzene, 3.51% 4-biphenyl formalin, 3.29% 4-biphenyl formic acid, 2.81% alkyl benzoic acid, and 3.7% new products. Compared to pure PET, the obvious changes in the content of the pyrolysis products were mainly reflected in the raised amounts of benzoic acid, formic acid, and 4- acetate benzoic acid, as well as in the decline in amounts of benzene, 4- biphenyl formalin, biphenyl, terphenyl, and alkyl benzoic acid.

Benzene acid was the main pyrolysis product of PET, and the raised contents indicated that the introduction of the SiMP-CMSs could prevent degradation of the PET. Benzene was a deep degradation product of benzene acid by a deacidification reaction [25], and the decline in benzene demonstrated that SiMP-CMSs could be restrained from deep degradation by benzene acid. In addition, aldehyde groups also accounted as the main pyrolysis products of the olefinic end groups, where biphenyl groups were generated during the PET free radical degradation process [26]. The decline in 4-biphenyl formalin and alkyl benzoic acid demonstrated that deep degradation of the olefinic end groups was prevented. The restraint in benzene acid and olefinic end groups from deep degradation could explain the improvement in thermal stability of the SiMP-CMSs/PET composites. Moreover, the decrease in biphenyl and terphenyl indicated that the PET free radical degradation process was slowed down by the presence of SiMP-CMSs. High amounts of formic acid could generate more CO_2_ and H_2_O during combustion, which may be the reason for the improved properties of LOI and UL-94 tests.

In addition to the above pyrolysis products, about 3.7% of the newly formed pyrolysis products were formed, including benzonitrile, (4-cyanophenyl) oxygen-containing acetic acid, acridine, 4-phenyl-2-azetidinone, and *N*-[1-(ethoxyl)]ethyl] phthalic acid. These products were mainly issued from the gasification and decomposition of MEL generated amino groups and triazine radicals, which, respectively, reacted with carboxylic acid and phenyl from the PET pyrolysis reaction to form nitriles, ammonia acid, an orthophenanthroline ring, and a nitrogen heterocyclic ring. These products played an important role in inhibiting deacidification and free radical reactions, thereby catalyzing the PET to char. Compounds containing phosphorous and silicon atoms were not detected and remained in the char residue.

On the other hand, the changes in the chemical structures of PET composites were explored by FTIR at different temperatures in the air atmosphere (Figure 9). At 400 °C, the characteristic absorbance peaks of the PET at 3427 cm^−1^ (stretching vibration of O–H bond), 2926 cm^−1^ and 2858 cm^−1^ (stretching vibration of C–H bond), 1651cm^−1^ (C=O stretching vibration), 1398 cm^−1^ (esters vibration absorption peak), and 1058 cm^−1^ (ketone vibration absorption) were visible, demonstrating the breakage of long polyester chains and ester groups. At 450 °C, the vibration absorption peak of the benzene ring appeared at 1525 cm^−1^, indicating that the products further decomposed to generate small molecule substances. At a high temperature of 500 °C, the aromatized char formed with part of the PET matrix crosslinked to the char.

Compared to the neat PET, the SiMP-CMSs/PET composites at 400 °C presented an ester group vibration absorption peak at 1739 cm^−1^. Also, the stretching and deformation of phenyl C=C in the plane were presented, respectively, at 1539 cm^−1^ and 1516 cm^−1^ [27], indicating that the SiMP-CMSs/PET delayed decomposition. Moreover, two new peaks occurred at 1286 cm^−1^ and 1086 cm^−1^, corresponding, respectively, to the stretching vibration of the P=O and P-O-C bonds [21]. Also, the weak absorption peak of C=N appeared at 1683 cm^−1^ [20]. These data demonstrate that the SiMP-CMSs firstly decomposed, and then phosphoric acid acted as an acid catalyst to promote the formation of a protective carbon layer with a P–O–C bond. Next, the melamine decomposed to form the crosslinked structure with the C=N structure, confirming the presence of some new nitrogenous pyrolysis products in 3.8. At 450 °C, the vibration peak of the aromatized structure at 1539 cm^−1^ and 1516 cm^−1^ gained in strength, and the two new peaks at 1286 cm^−1^ and 1170 cm^−1^ were caused by the stretching vibration of the P=O bond. Moreover, the antisymmetric vibration peak of the P–O–C bond was presented at 1124 cm^−1^. These changes indicate the promotion of the PET matrix by the SiMP-CMSs for forming a P-O-C char structure. At temperatures higher than 500 °C, the C=N peak at 1683 cm^−1^ almost vanished, implying that most MEL gasified and decomposed, and the products were released into the gas phase. These data confirm the importance of the SiMP-CMSs’ function in flame inhibition and prove the occurrence of new nitrogenous materials during Py-GC-MS testing.

As the thermal temperature further rose to 550 °C or 600 °C, the cross-linked char structure at 1516 cm^−1^ remained unchanged, but the vibration intensity became higher than that of pure PET, explaining the reason for the higher thermal oxygen stability of the char layer formed by the SiMP-CMSs/PET when compared to that formed by pure PET.

### 3.5. Char Structure, Flame-Retardant, and Smoke Suppression Mechanism

To further investigate the flame-retardant mechanism, the char structure after cone testing was analyzed by SEM, and the results are depicted in Figure 10. The mass fraction of the SiMP-CMSs was 3%. The char structure of the neat PET present thin fragments with almost no barrier effect (Figure 10a). By comparison, the char structures of the SiMP-CMSs/PET composites (the mass fraction of the SiMP-CMSs was 3%) appear as relatively complete barrier layers with certain thicknesses. In other words, the amount of char residue increased, indicating that the SiMP-CMSs could catalyze the PET itself to form the char layer. In addition, EDS data in the upper right corner also reveal the presence of C, O, N, P, and Si elements in the char residue. The existence of a P element again demonstrates that phosphoric acid plays an important role in the char layer. The existence of the N element confirms that, despite gasification of the MEL, the triazine ring structure remained existent in the char residue as the FTIR results showed, explaining the reason for the formation of the crosslinked char.

Taking into account the above analyses, the flame retardant mechanism of SiMP-CMSs for PET is mainly associated with the three aspects shown in Figure 11. First, CMSs played a role in the heat insulation effect, partially catalyzing the char effect by vinyl formate (decomposition products of the CMSs themselves). Second, the SiMP present on the surface of the CMSs decomposed phosphoric acid and melamine, where the phosphoric acid promoted the char layer of the PET in the condensed phase and reduced the matrix amount to producing smoke. The P=O bond deactivated the char layer and increased its barrier properties, cutting off the release of smoke. Third, the melamine vaporized and covered the upper polymer surface, and then decomposed to generate ammonia and nitrogenous gaseous products that diluted the O_2_ concentration and inhibited the flame by blocking the way of smoke suppression. The triazine ring structure existed in the char structure by promoting the cross-link of the PET, which improved the thermal-oxidative stability of the char and lowered the smoke amounts. These three aspects ultimately made the PET matrix simultaneously achieve flame retardancy and smoke suppression.

### 3.6. Tensile Strength

The mechanical properties would be ranked as basic requirements for functionalized materials, as well an important performance requirement of flame retardant materials. The modification of MP-CMSs by APTS aimed to improve the interface binding between MP-CMSs and the PET matrix. The tensile strength of the MP-CMSs/PET and the SiMP-CMSs/PET composites were tested to examine the interface state, and the results are gathered in Figure 12. Obviously, the addition of unmodified MP-CMSs caused serious deterioration in the tensile strength of the PET. At 0.5%, the tensile strength of the MP-CMSs/PET composites decreased to nearly 22.7%. As the MP-CMSs rose, the tensile strength decreased to nearly 52.8%. This decrease was mainly due to the weak interface binding and different polarities between the MP-CMSs particles and PET matrix, resulting in and ineffective transfer of bearing stress. By contrast, the tensile strength of the SiMP-CMSs/PET composites was higher than that of the MP-CMSs/PET, indicating that the outer coupling agent strengthened the interface binding between the MP-CMSs particles and the PET matrix, and the interface buffer layer formed between the SiMP-CMSs particles and the PET matrix. Adding 0.5%, 1%, and 2% SiMP-CMSs, the tensile strength of the SiMP-CMSs/PET composites was estimated to be 49.65 MPa, 48.16 MPa, and 47.69 MPa, respectively. These corresponded to decreases of 2.84%, 5.75%, and 6.67%, which did not affect the usage requirements. However, the increase of the SiMP-CMSs to 3% reduced the tensile strength by 20.3%, slightly affecting the usage requirements.

## 4. Conclusions

Integrated composite flame retardant MP-CMSs were successfully prepared by a solvent hydrothermal method, then modified by a coupling agent APTS to form novel SiMP-CMSs with acid, carbon, and gas sources. The SiMP-CMSs showed excellent flame retardancy and smoke suppression for PET. The addition of only 3% SiMP-CMSs rose the LOI value of the PET from 21% ± 0.1% to 27.7% ± 0.3%, reaching the V-0 burning rate. The SiMP-CMSs did not only reduce the heat damage caused by PET combustion, but also inhibited the release of smoke. The pk-HRR value reduced from 513.22 to 221.66 kW/m^2^, and the smoke parameter value decreased from 229830.18 kW/kg to 81892.29 kW/kg. The fire performance index rose from 0.0740 to 0.1714 m^2^s/kW, demonstrating that the SiMP-CMSs had an obviously reduced fire risk. The proportion of three flame retardant modes, physical barrier, flame inhibition, and char effects, was estimated to be 44.53%, 19.04%, and 9.04%, respectively. Also, the tensile strength of the SiMP-CMSs/PET composites decreased only by 20.3%, satisfying the usage requirements.

## Figures and Tables

**Figure 1 polymers-11-00545-f001:**
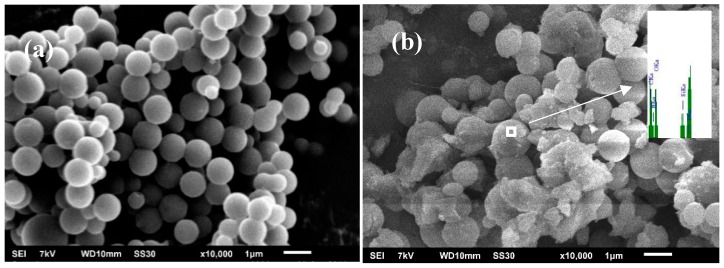
Scanning electron microscopy (SEM) and energy dispersive spectroscopy (EDS) images of carbon microspheres (CMSs) (**a**) and silane melamine phosphate carbon microspheres (SiMP-CMSs) (**b**).

**Figure 2 polymers-11-00545-f002:**
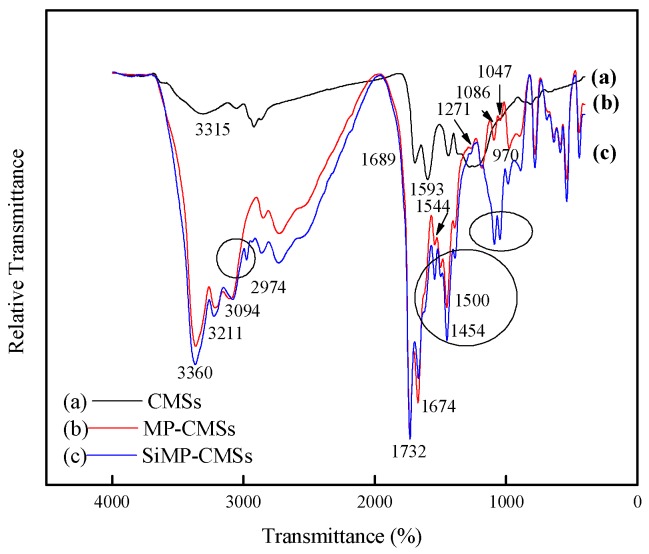
Fourier transformation infrared (FTIR) spectra of CMSs (**a**), MP-CMSs (**b**) and SiMP-CMSs (**c**).

**Figure 3 polymers-11-00545-f003:**
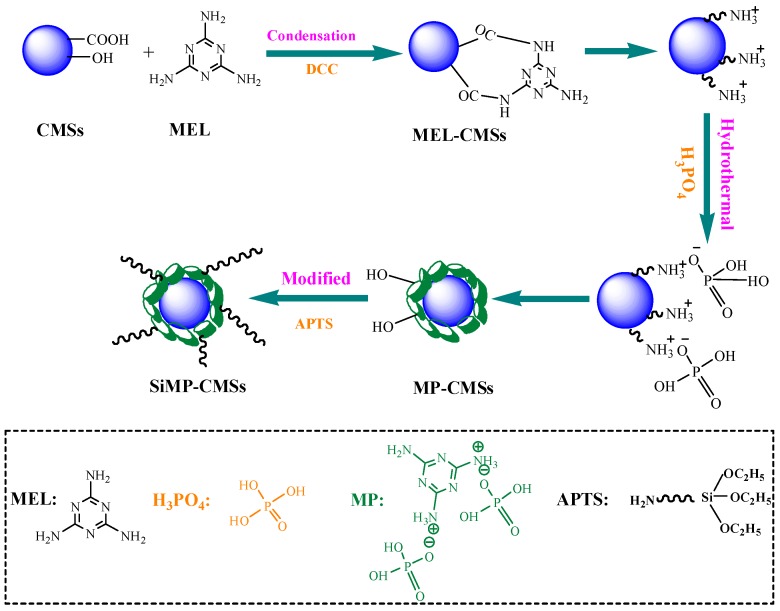
The formation schematic process of SiMP-CMSs. APTS, 3-Aminopropyltriethoxysilane; MEL, melamine.

**Figure 4 polymers-11-00545-f004:**
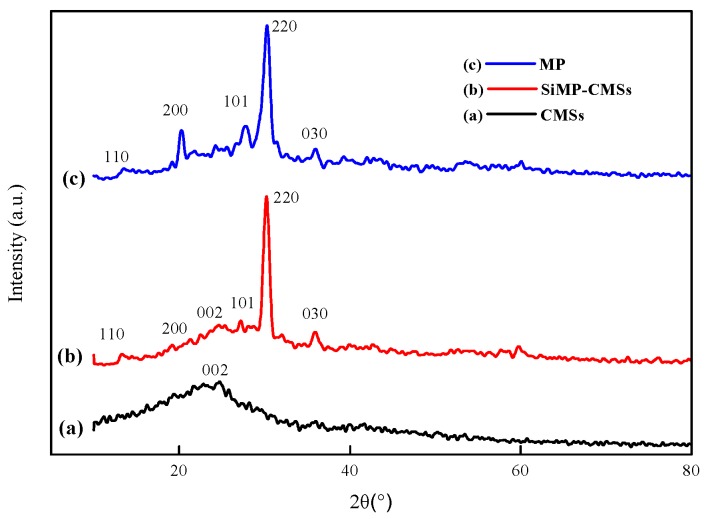
X-Ray diffraction (XRD) spectrum of CMSs (**a**), SiMP-CMSs (**b**) and MP (**c**).

**Figure 5 polymers-11-00545-f005:**
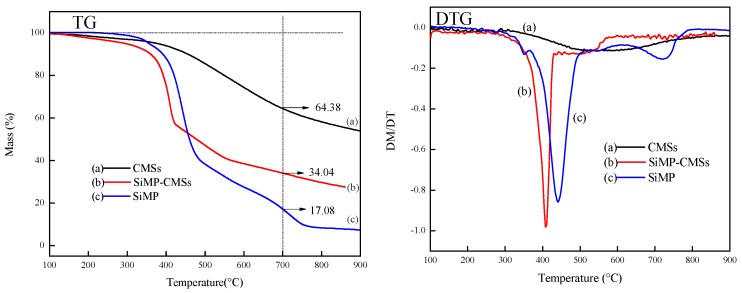
TG and DTG curves of CMSs (**a**), SiMP-CMSs (**b**), and SiMP (**c**).

**Figure 6 polymers-11-00545-f006:**
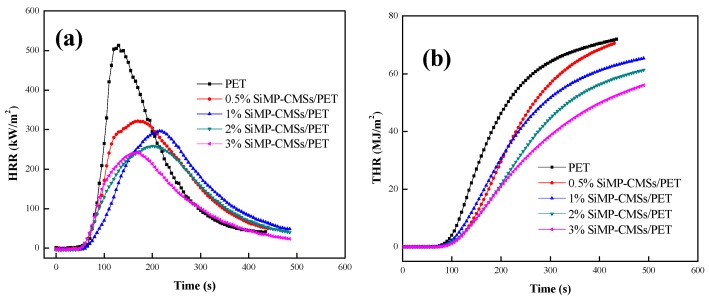
Heat release rate (HRR) (**a**) and total heat release (THR) (**b**) curves of SiMP-CMSs/ Poly(ethylene terephthalate (PET) composites.

**Figure 7 polymers-11-00545-f007:**
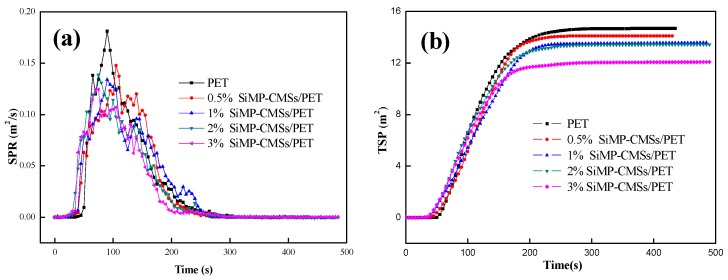
Smoke production rate (SPR) (**a**) and total smoke production (TSP) (**b**) curves of the SiMP-CMSs/PET composites.

**Figure 8 polymers-11-00545-f008:**
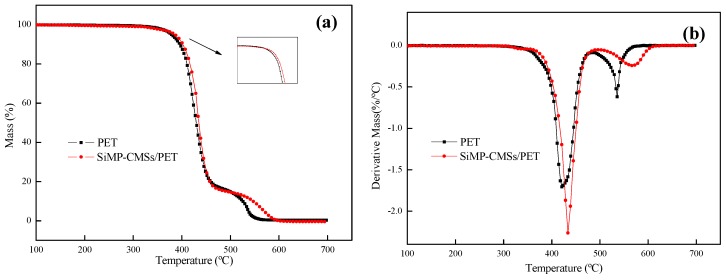
TGA (**a**) and DTG (**b**) curves of the SiMP-CMSs/PET composites in the air atmosphere.

**Figure 9 polymers-11-00545-f009:**
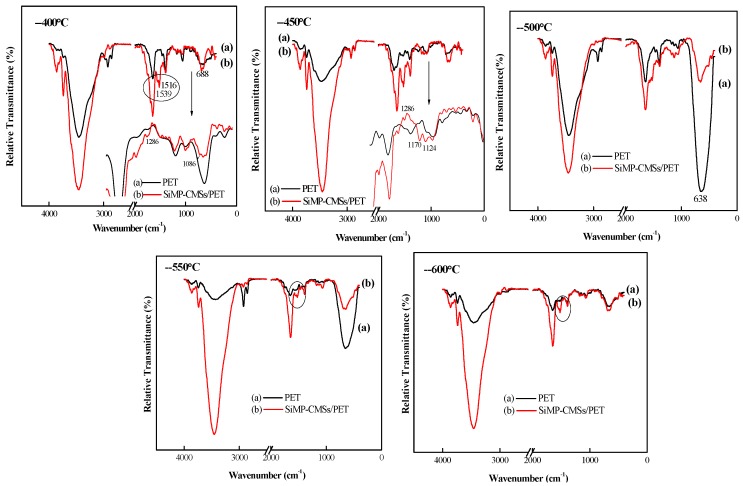
FTIR spectra of the thermal oxidative degradation product of the SiMP-CMSs/PET composites at different temperatures.

**Figure 10 polymers-11-00545-f010:**
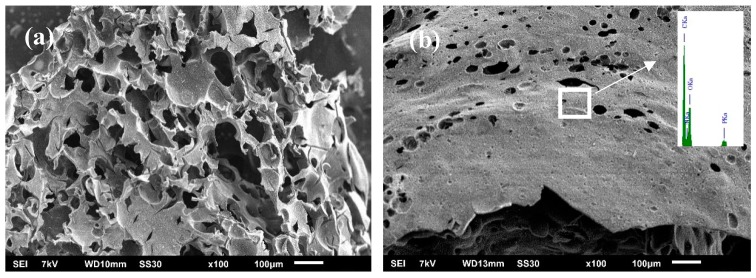
Char residue SEM image of PET (**a**) and SiMP-CMSs/PET composites (**b**).

**Figure 11 polymers-11-00545-f011:**
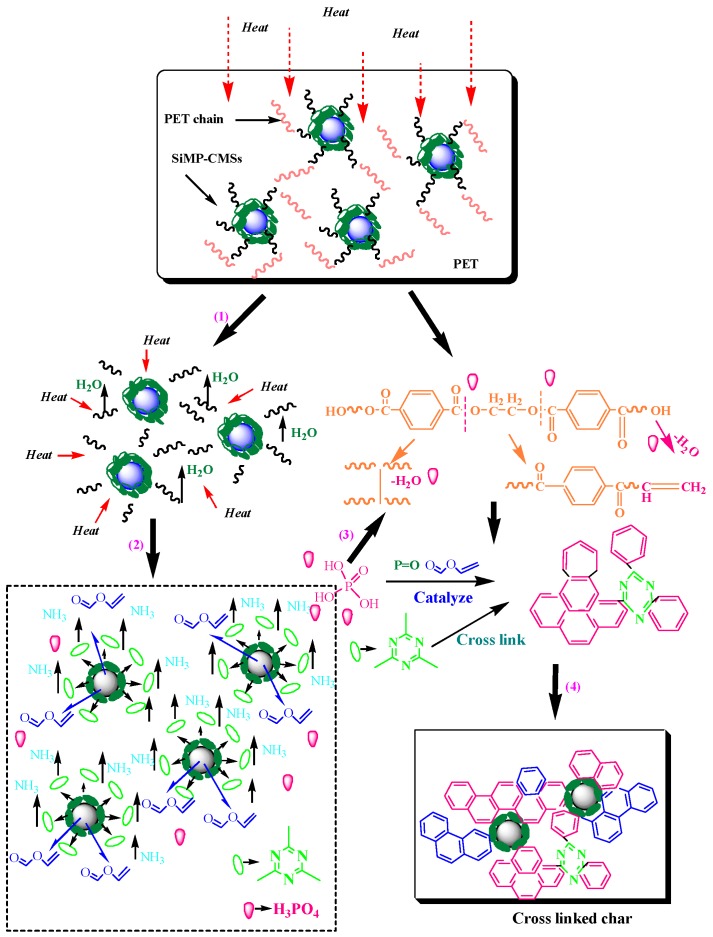
Process schematic diagram of the SiMP-CMSs flame retardant PET.

**Figure 12 polymers-11-00545-f012:**
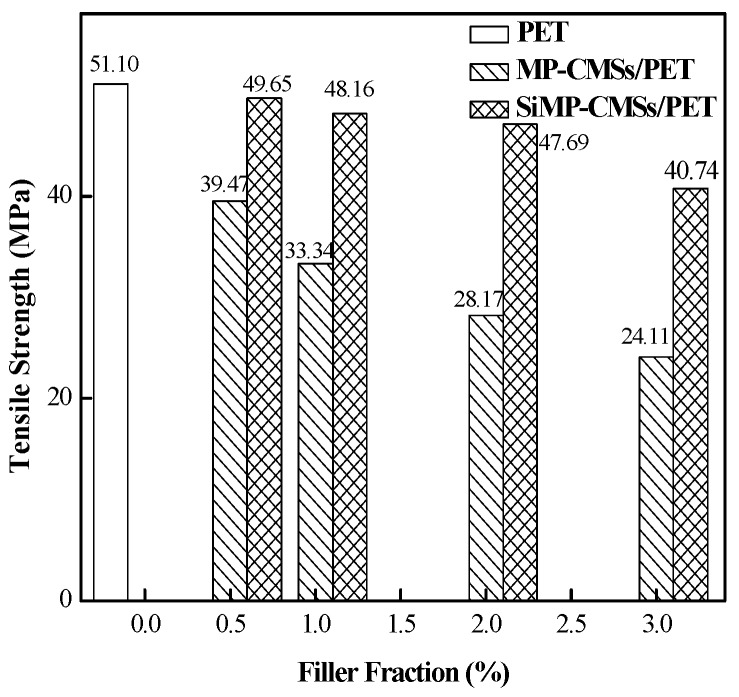
Tensile Strength of the SiMP-CMSs/PET composites.

**Table 1 polymers-11-00545-t001:** Limiting oxygen index (LOI) values and UL-94 rating of SiMP-CMSs/PET composites.

Sample	LOI (%)	*T*_1_/*T*_2_ (s) ^a^	*T*_f_ (s) ^b^	Igniting Cotton	UL-94 Rating
PET	21.0 ± 0.1	11.9/7.2	85.5	Yes	NR
0.5%SiMP-CMSs/PET	25.8 ± 0.2	6.9/4.4	56.5	Yes	V-2
1%SiMP-CMSs/PET	26.3 ± 0.1	5.7/3.6	46.5	Yes	V-2
2%SiMP-CMSs/PET	27.1 ± 0.2	3.8/2.4	31.0	No	V-0
3%SiMP-CMSs/PET	27.7 ± 0.3	3.0/1.7	23.5	No	V-0

^a^ Average combustion times after the first and second applications of the flame, ^b^ The total duration (five specimens) of flaming.

**Table 2 polymers-11-00545-t002:** Main heat parameters of SiMP-CMSs/PET composites. MEHC, mean effective heat of combustion; FPI, fire performance index.

Sample	pk-HRR (kW/m^2^)	THR (MJ/m^2^)	MEHC (MJ/kg)	FPI (m^2^s/kW)	Residue (%)
PET	513.2	72.0	23.0	0.07	11.8
0.5%SiMP-CMSs/PET	320.2	70.5	22.3	0.12	13.8
1%SiMP-CMSs/PET	296.2	65.3	20.5	0.13	15.2
2%SiMP-CMSs/PET	258.8	61.3	19.9	0.15	17.4
3%SiMP-CMSs/PET	221.7	56.0	18.6	0.17	19.8

**Table 3 polymers-11-00545-t003:** Smoke parameters of the SiMP-CMSs/PET composites. SEA, specific extinction area; SP, smoke production.

Sample	TSP (m^2^)	av-SEA (m^2^/kg)	SP (kW/kg)	CO Yield (kg/kg)	CO_2_ Yield (kg/kg)
PET	14.7	447.8	229,830.2	0.06	1.36
0.5%SiMP-CMSs/PET	14.1	425.5	136,256.8	0.06	1.33
1%SiMP-CMSs/PET	13.6	406.2	120,297.6	0.06	1.12
2%SiMP-CMSs/PET	13.4	379.8	98,274.5	0.05	1.07
3%SiMP-CMSs/PET	12.1	369.5	81,892.3	0.04	1.03

**Table 4 polymers-11-00545-t004:** Quantitative comparison of flame-retardant modes for SiMP-CMSs/PET composites.

Samples		Barrier Effect (%)	Flame Inhibition Effect (%)	Charring Effect (%)
SiMP-CMSs/PET	0.5%	36.35	3.04	2.25
	1%	36.43	10.87	3.83
	2%	40.80	13.47	6.81
	3%	44.53	19.04	9.04

**Table 5 polymers-11-00545-t005:** Thermal stability and degradation of the SiMP-CMSs/PET composites.

Sample	*T*_onest_^a^ (°C)	*T*_max1_^b^ (°C)	*T*_max2_^c^ (°C)
PET	378.6	421.1	535.6
SiMP-CMSs/PET	387.7	433.5	572.3

^a^ The initial decomposition temperature, ^b,c^
*T*_max_ at the first and second decomposition stage.

**Table 6 polymers-11-00545-t006:** Main pyrolysis products of SiMP-CMSs/PET composites.

Peak No.	Main Pyrolysis Product	Content (%)	
		PET	SiMP-CMSs/PET
1	Formic acid CH_2_O_2_	3.03	9.02
2	Acetic acid C_2_H_4_O_2_	4.10	---
3	Propanediol C_3_H_8_O_2_	1.94	0.65
4	Benzene C_6_H_6_	7.23	5.17
5	BenzonitrileC_6_H_5_	---	0.77
6	Phenylglyoxal C_8_H_6_O_2_	7.25	---
7	Benzoyl methyl ketone C_9_H_10_O_2_	---	7.11
8	Benzene acid C_7_H_6_O_2_	34.12	38.57
9	2,2-diphenylethenone C_12_H_14_O	1.00	1.13
10	Biphenyl C_12_H_10_	4.39	3.15
11	(4-cyanophenyl) oxygen-containing acetic acid C_9_H_5_NO_3_	---	1.09
12	4-Carboxybenzaldehyde C_8_H_5_O_3_	8.05	2.84
13	Diethyl phthalate C_12_H_12_O_4_	---	1.10
14	4-acetylbenzoic acid C_9_H_8_O_3_	4.15	5.84
15	Acridine C_13_H_9_N	---	0.16
16	9-Fluorenone C_13_H_8_O	0.82	0.55
17	4-biphenyl formalin C_13_H_10_O	5.52	3.51
18	4-biphenyl formic acid C_13_H_10_O_2_	2.64	3.29
19	diethylene glycol dibenzoate C_16_H_14_O_4_	10.90	9.89
20	Isopropyl phenyl ketone C_10_H_12_O	1.32	0.72
21	Vinyl benzoate C_9_H_8_O_2_	---	1.02
22	4-phenyl-2-azetidinone C_9_H_9_NO	--	0.53
23	*N*-[1-(ethoxyl)] ethyl] phthalic acid C_13_H_15_NO_5_	---	1.15
…	…	…	…

“...” represents the amount of some products was less than 0.5%.

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
