# Peer review of "Construction of Carbon Microspheres-Based Silane Melamine Phosphate Hybrids for Flame Retardant Poly(ethylene Terephthalate)"

_polymers, 2019, doi:10.3390/polym11030545_

Round 1

Reviewer 1 Report

Please find my comments on file enclosed

Author Response

Dear reviewer,

Thanks for your letter and the comments on our manuscript entitled “In situ construction of integrated carbon microspheres based silanated melamine polyphosphate hybrids and effects on flame retardancy and smoke suppression of poly(ethylene terephthalate)”(ID: polymers-461942).

Those comments are very helpful for revising and improving our paper, as well as the important guiding significance to other research. We have studied the comments carefully and made corrections which we hope meet with approval. The main corrections are marked in red font in the manuscript and the responds to the reviewer’ comments are as follows.

Reply to the reviewer:

Q1. MEL corresponds to melamine not to melamine polyphosphate. Moreover MP corresponds to melamine phosphate. Please clarify your abbreviations.

Response: Thank you for the reviewer’s remind. We agreed with the advice and clarified the abbreviations and revised the similar mistake in the paper.

Q2. Give the detail of the abbreviation of APTS.

Response: According to the reviewer’s advice, we have supplemented the abbreviation of APTS in the revised paper.

Q3. The number of decimal places is too important compared to the actual accuracy of the measurement.

Response: We are grateful for the reviewer’s suggestion. And another reviewer also mentioned the number of decimal places was too many. Consideration with the comments from the reviewers, we have revised the number of decimal places in the paper.

Q4. It must be rather "pentaerythritol".

Response: Thanks again for the reviewer’s advice. The detail of the PER abbreviation was pentaerythritol. And we have corrected it in the revised paper.

Q5. It was not clear : "with existent synergistic flame retardancy between ..."

Response: We are thankful for the reviewer’s suggestion. To express the meaning clearly, we have supplemented the detail and added the statement “such as the synergistic effect of expandable graphite on the melamine polyphosphate/dipentaerythritol/polypropylene (PP) [8], multiwalled carbon nanotubes functionalized with IFR on improving the flame retardant of ABS matrix[9], thermally reduced graphite oxide strengthening the residue structure of ammonium polyphosphate /magnesium hydroxide/PP[10], et al.” in the introduction.

Q6. “ …then added into a two-screw extruder (CET35–40D)”. supplier ? city ? country ?

Response: Thank you for the reviewer’s remind. We have added the information about the supplier, city and country of the two -screw extruder (CET35–40D).

Q7. The tensile strength test must be presented (apparatus, experimental conditions).

Response: Thank you for the reviewer’s remind. We have supplemented the tensile strength test condition. The statement “The tensile test was carried out by a universal material testing system (model CMT6104, Shenzhen, China) at room temperature with gauge length of 30 mm and crosshead speed of 50 mm/min.” have been added in the revised paper.

Q8. All parameters calculated from cone results must be presented. In particularly, you have to give the calculi for MEHC and FPI Idem for smoke parameters.

Response: We are thankful for the reviewer’s advice and suggestion. We have supplemented all parameters in the material characterization and gave the formula of FPI and SP. The statement “ The time to ignition (TTI), total heat release (THR), mean effective heat of combustion (MEHC), smoke production rate (SPR) total smoke production (TSP) and mean specific extinction area (av-SEA) parameters were directly obtained from the cone results. And fire performance index (FPI) (FPI=TTI/pk-HRR) and the smoke parameter (SP) (SP=av-SEA*pk-HRR) were calculated from the cone results.” have been added in the revised paper.

Q9. It was SiMP not MP. Could you explain why the first main degradation of SiMP-CMSs is shifted around 20 °C below than that of SiMP ? The authors should discuss on the final residue for the three elements. Indeed, even if some interactions occur between CMSs and SiMP, the final residue of SiMP-CMSs could give an indication on the mass ratio between SiMP and CMSs in SiMP-CMSs.

Response: Thanks again for the reviewer’s valuable advice and question. It was SiMP and we have corrected it. The reason for “the first main degradation of SiMP-CMSs was shifted around 20°C below than that of SiMP” may be that the decomposition of unstable reaction products on the SiMP-CMSs surface. Consideration with the reviewer’s advice, we have calculated the mass ratio between SiMP and CMSs in SiMP-CMSs according to the final residue at 700°C.

Q10. please check the actual accuracy of the measurements.

Response: Thank you again for the reviewer’s advice. We have checked and revised in the paper.

Q11. “namely β-methene” was “methylene”?

Response: Yes. It was β-methylene. We have checked the paper carefully and revised similar mistake in the paper.

We tried our best to improve the manuscript and made some changes in the manuscript. All revised portions are marked in red font in the revised manuscript which we hope to meet with approval.

Once again, thank you very much for your constructive comments and suggestions which would help us to improve the quality of the paper. The revised paper has been upload.

Kind regards.

Baoxia Xue

Email: xuebaoxia@tyut.edu.cn

Reviewer 2 Report

The paper from Xue et al reports on the synthesis of carbon microspheres-based silanated melamine polyphosphate hybrids and their us as flame retardants and smoke suppressants for pET. Though the proposed idea seems to show some novelty, the manuscript requires further experimental activity.

Some comments and suggestions are listed as follows:

- Figure 5: please add the dTG curves

- Abstract and Table 3: too many decimal digits for pkHRR and THR. Furthermore, please add the final residue and CO/CO2 yield to Table 3

- Table 4: too many decimal digits for all the parameters

- it could be reasonable to perform SEM-EDX analyses first on the flame retarded compounds, in order to assess the homogeneity of distribution of the FR; then, the same analyses should be performed on the residues after cone calorimetry tests

- what about the mechanical behavior of the flame retarded compounds?

Author Response

Dear reviewer,

Thanks for your letter and the comments on our manuscript entitled “In Situ Construction of Integrated Carbon Microspheres Based Silanated Melamine Polyphosphate Hybrids and Effects on Flame Retardancy and Smoke Suppression of Poly(ethylene terephthalate)”(ID: polymers-461942).

Those comments are very helpful for revising and improving our paper, as well as the important guiding significance to other research. We have studied the comments carefully and made corrections which we hope meet with approval. The main corrections are marked by red lines in the manuscript and the responds to the reviewer’ comments are as follows.

“The paper from Xue et al reports on the synthesis of carbon microspheres-based silanated melamine polyphosphate hybrids and their us as flame retardants and smoke suppressants for pET. Though the proposed idea seems to show some novelty, the manuscript requires further experimental activity. Some comments and suggestions are listed as follows”

Reply to the reviewer:

Q1. Figure 5: please add the dTG curves

Response: Thank you for the reviewer’s advice. We have added the DTG curves in the Figure 5 in the revised paper.

Q2. Abstract and Table 3: too many decimal digits for pkHRR and THR. Furthermore, please add the final residue and CO/CO2 yield to Table 3

Response: We are thankful for the reviewer’s suggestions. We have reduced the decimal digits for the pkHRR, THR, and other parameters in the whole paper. In addition, we have supplemented the final residue and CO/CO2 yield to Table. The revised part have been marked in red font.

Q3. Table 4: too many decimal digits for all the parameters

Response: According to the reviewer’s advice. We have revised these in the paper.

Q4. it could be reasonable to perform SEM-EDX analyses first on the flame retarded compounds, in order to assess the homogeneity of distribution of the FR; then, the same analyses should be performed on the residues after cone calorimetry tests

Response: We are grateful for the reviewer’s suggestions. We agreed with the idea. In fact, we have proved the elements of residues by SEM-EDs analyses in Section 3.5 of the paper.

Q5. what about the mechanical behavior of the flame retarded compounds?

Response: Thanks again for the reviewer’s question. We have tested the tensile strength of the flame retarded compounds to evaluate the mechanical behavior. Adding 0.5%, 1% and 2% SiMP-CMSs, the tensile strength of SiMP-CMSs/PET composites were estimated to 49.65MPa, 48.16MPa and 47.69MPa, respectively. These corresponded to decreases of 2.84%, 5.75% and 6.67%, which did not affect the usage requirements. However, the increase of SiMP-CMSs to 3% reduced the tensile strength by 20.3%, slightly affecting the usage requirements.

We tried our best to improve the manuscript and made some changes in the manuscript. All revised portions are marked in red font in the revised manuscript which we hope to meet with approval.

Once again, thank you very much for your constructive comments and suggestions which would help us to improve the quality of the paper. The revised paper has been upload.

Kind regards.

Baoxia Xue

Email: xuebaoxia@tyut.edu.cn

Reviewer 3 Report

Review report: Manuscript ID: polymers-461942

In this manuscript “In Situ Construction of Integrated Carbon Microspheres Based Silanated Melamine Polyphosphate Hybrids and Effects on Flame Retardancy and Smoke Suppression of Poly(ethylene terephthalate)” the author prepared a hybrid material based on melamine polyphosphates and carbon microspheres which could be used as a potential flame retardant material for polyethylene terepthalate. The work is of interest to the audience but has not organized properly based on my below given comments. If the author modify the paper based on the below given comments then it may be accepted for publications. The comments are given below:

Q1. The title of the papers seems too long and does not give a instant objective of the work of what it is. I would suggest shortening the title with a precise one.

Q2. The abstract seems a bit longer. Need to be shortened with important findings. All Acronyms should be defined at its first instance for better understanding of the reader. For example APTS, SiMPCMSs etc

Q3. This is quite obvious of a statement in abstract.

 “The morphological observations, Fourier-transform Infrared spectrometry (FTIR) and 26 thermogravimetry (TG) tests indicated that SiMP layer was loaded on the CMS surface” Hence not needed.

Q4. How did you find out that smoke release rate inhibited. How the FPI calculated ?

Q5. Need a reference for this statement in introduction section. “However, the traditional carbon source 51 existed in defects of low molecular weight, which could migrate easily to deteriorate the mechanical 52 properties of polymers.”

Q6. In the introduction section, the literature reference related to improve flame retardency is not enough. The author need to put more rent work that has been done and say whats so new about their current work. The author may cite and use this reference in introduction section. DOI: 10.1016/j.jcis.2013.04.053; DOI.org/10.1016/j.compositesb.2012.04.028; DOI:10.1016/j.polymdegradstab.2011.03.020

Q7. Acronym AR stand for analytical reagent. Though it was clear but need to be defined at its first instance.

Q8. PET chips: instead of saying chips, pellets would be a good choice of name

Q9. Why PET pellets dried in vaccumm oven ? I understand it is to remove moisture from starting material but that need to be mentioned.

Q10. Please mention the page number.

Q11. 2.3 Preparation of SiMP-CMSs/PET composites: In this section the author mentioned the mechanical properties are measured by injection molding. How the mechanical properties measured by injection molding ? May be author tried to say the samples are prepared by injection molding for mechanical measurement ?. Please modify .

Q12. How the specimen SiMP-CMSs were made for TEM ?

Q13. Why fig 1b looks more noisy compared to fig 1 a. From the EDX analysis, the author found the outer layer made of Si-MP. This is absurd.

Q14. Table 1 is of little use since the author has already explained it in the text.

Q15. How many times the LOI measured for each sample for consistency. It should be represented with an error value plus/or minus.

Q16. In the characterization section, author said morphology was characterized by SEM and TEM but I have not found any results of TEM.

Q17. For tensile strength the author need to put the stress strain graph as well along with the histogram.

Author Response

Dear reviewer,

Thanks for your letter and the comments on our manuscript entitled “In Situ Construction of Integrated Carbon Microspheres Based Silanated Melamine Polyphosphate Hybrids and Effects on Flame Retardancy and Smoke Suppression of Poly(ethylene terephthalate)”(ID: polymers-461942).

Those comments are very helpful for revising and improving our paper, as well as the important guiding significance to other research. We have studied the comments carefully and made corrections which we hope meet with approval. The main corrections are marked in red font in the manuscript and the responds to the reviewer’ comments are as follows.

“In this manuscript “In Situ Construction of Integrated Carbon Microspheres Based Silanated Melamine Polyphosphate Hybrids and Effects on Flame Retardancy and Smoke Suppression of Poly(ethylene terephthalate)” the author prepared a hybrid material based on melamine polyphosphates and carbon microspheres which could be used as a potential flame retardant material for polyethylene terepthalate. The work is of interest to the audience but has not organized properly based on my below given comments. If the author modified the paper based on the below given comments then it may be accepted for publications. The comments are given below:”

Reply to the reviewer:

Q1. The title of the papers seems too long and does not give a instant objective of the work of what it is. I would suggest shortening the title with a precise one.

Response: We are grateful for the reviewer’s advice. We have shortened the title.

Q2. The abstract seems a bit longer. Need to be shortened with important findings. All Acronyms should be defined at its first instance for better understanding of the reader. For example, APTS, SiMP-CMSs etc

Response: Thanks for the reviewer’s valuable advice. We have shortened the abstract part and also supplemented the acronyms for better understanding of the reader.

Q3. This is quite obvious of a statement in abstract. “The morphological observations, Fourier-transform Infrared spectrometry (FTIR) and 26 thermogravimetry (TG) tests indicated that SiMP layer was loaded on the CMS surface” Hence not needed.

Response: According to the reviewer’s suggestion, we have deleted the above statement and revised in the paper.

Q4. How did you find out that smoke release rate inhibited. How the FPI calculated ?

Response: Thank you for the reviewer’s question. According to the smoke parameters, the av-SEA, TSP and SP in Table 3 and the smoke release rate curve in Figure 7, these parameters all declined, which implied the smoke release rate have been inhibited. For the calculation of FPI, we have supplemented the formula of FPI in the revised paper. FPI=TTI/pk-HRR.

Q5. Need a reference for this statement in introduction section. “However, the traditional carbon source 51 existed in defects of low molecular weight, which could migrate easily to deteriorate the mechanical 52 properties of polymers.”

Response: Thank you for the reviewer’s remind. We have added the reference for this statement in introduction section.

Q6. In the introduction section, the literature reference related to improve flame retardency is not enough. The author need to put more rent work that has been done and say whats so new about their current work. The author may cite and use this reference in introduction section. DOI: 10.1016/j.jcis.2013.04.053; DOI.org/10.1016/j.compositesb.2012.04.028; DOI:10.1016/j.polymdegradstab.2011.03.020

Response: We are thankful for the reviewer’s suggestion. We have revised the introduction section. According to the above recommended reference, we have studied and knew the TiO2 played the key role in improving the thermal stability of polymers. In another paper, we will cited and use these reference. Thank you for the reviewer’s understanding.

Q7. Acronym AR stand for analytical reagent. Though it was clear but need to be defined at its first instance. Q8. PET chips: instead of saying chips, pellets would be a good choice of name.Q9. Why PET pellets dried in vaccumm oven? I understand it is to remove moisture from starting material but that need to be mentioned. Q10. Please mention the page number.

Response: We are grateful for the reviewer’s valuable suggestions. According to the advice, we have supplemented the above information by item in the revised paper.

Q11. 2.3 Preparation of SiMP-CMSs/PET composites: In this section the author mentioned the mechanical properties are measured by injection molding. How the mechanical properties measured by injection molding? May be author tried to say the samples are prepared by injection molding for mechanical measurement? Please modify.

Response: Thanks for the reviewer’s careful reading. This is a mistake. In fact, we expressed the samples are prepared by injection molding. We have modified this in paper.

Q12. How the specimen SiMP-CMSs were made for TEM?

Response: This is a failure to express. In fact, we have not tested the SiMP-CMSs by TEM method.

Q13. Why fig 1b looks more noisy compared to fig 1 a. From the EDX analysis, the author found the outer layer made of Si-MP. This is absurd.

Response: Thanks for the reviewer’s question. The fig 1b and fig 1a were the pure and modified CMSs. Due to the different shooting angle and sample heterogeneity, they seemed to be different. And the expression “the outer layer made of SiMP” was not accurate. We agreed with the reviewer’s advice. We have modified the expression. The EDX analysis result only proved the SiMP may exist on the surface.

Q14. Table 1 is of little use since the author has already explained it in the text.

Response: According to the reviewer’s advice, we have deleted the Table 1.

Q15. How many times the LOI measured for each sample for consistency. It should be represented with an error value plus/or minus.

Response: We are grateful for the reviewer’s question and advice. In fact, we have tested 10 times LOI measured for each sample. And we have supplemented the plus/or minus in paper.

Q16. In the characterization section, author said morphology was characterized by SEM and TEM but I have not found any results of TEM.

Response: Thank you for the reviewer’s careful suggestion. This is a failure to express. We have corrected it.

Q17. For tensile strength the author need to put the stress strain graph as well along with the histogram.

Response: Thank you for the reviewer’s advice. According to our situation, we can’t provide the stress graph. On one hand, for tensile strength, we also tested 10 times for one sample. And the average value of 10 samples was the final result. The stress graph of every sample was different. On other hand, our equipment can’t satisfy the test condition.

We tried our best to improve the manuscript and made some changes in the manuscript. All revised portions are marked in red font in the revised manuscript which we hope to meet with approval.

Once again, thank you very much for your constructive comments and suggestions which would help us to improve the quality of the paper.

Kind regards.

Baoxia Xue

Email: xuebaoxia@tyut.edu.cn

Round 2

Reviewer 2 Report

The manuscript has been revised according to the reviewer's comments and suggestions. Now, it seems suitable for publication in Polymers.